# PoxiPred: An Artificial-Intelligence-Based Method for the Prediction of Potential Antigens and Epitopes to Accelerate Vaccine Development Efforts against Poxviruses

**DOI:** 10.3390/biology13020125

**Published:** 2024-02-17

**Authors:** Gustavo Sganzerla Martinez, Mansi Dutt, David J. Kelvin, Anuj Kumar

**Affiliations:** 1Department of Microbiology and Immunology, Dalhousie University, Halifax, NS B3H 4H7, Canada; gustavo.sganzerla@dal.ca (G.S.M.); mansidutt@dal.ca (M.D.); kumaranuj@dal.ca (A.K.); 2Department of Pediatrics, Izaak Walton Killam (IWK) Health Center, Canadian Center for Vaccinology (CCfV), Halifax, NS B3H 4H7, Canada; 3Laboratory of Immunity, Shantou University Medical College, Shantou 512025, China; 4BioForge Canada Limited, Halifax, B3N3B9, NS, Canada

**Keywords:** *Poxviridae*, vaccine, epitopes, artificial intelligence, antigens

## Abstract

**Simple Summary:**

Poxviruses are large, complex, enveloped, double-stranded DNA viruses known to cause contagious diseases in humans and animals. There is a pressing and urgent demand for the development of effective vaccines to combat diverse sets of poxviruses. In the absence of significant methods, predicting T-cell epitopes and antigens for poxviruses remains a challenging task. In this study, we have employed an artificial intelligence-based approach and developed a method named PoxiPred for the prediction of proteome-wide antigens and T-cell epitopes. We anticipate that this open-source tool will be useful to the scientific community for accelerating vaccine development efforts against contagious poxviruses.

**Abstract:**

*Poxviridae* is a family of large, complex, enveloped, and double-stranded DNA viruses. The members of this family are ubiquitous and well known to cause contagious diseases in humans and other types of animals as well. Taxonomically, the *poxviridae* family is classified into two subfamilies, namely *Chordopoxvirinae* (affecting vertebrates) and *Entomopoxvirinae* (affecting insects). The members of the *Chordopoxvirinae* subfamily are further divided into 18 genera based on the genome architecture and evolutionary relationship. Of these 18 genera, four genera, namely Molluscipoxvirus, Orthopoxvirus, Parapoxvirus, and Yatapoxvirus, are known for infecting humans. Some of the popular members of *poxviridae* are variola virus, vaccine virus, Mpox (formerly known as monkeypox), cowpox, etc. There is still a pressing demand for the development of effective vaccines against poxviruses. Integrated immunoinformatics and artificial-intelligence (AI)-based methods have emerged as important approaches to design multi-epitope vaccines against contagious emerging infectious diseases. Despite significant progress in immunoinformatics and AI-based techniques, limited methods are available to predict the epitopes. In this study, we have proposed a unique method to predict the potential antigens and T-cell epitopes for multiple poxviruses. With PoxiPred, we developed an AI-based tool that was trained and tested with the antigens and epitopes of poxviruses. Our tool was able to locate 3191 antigen proteins from 25 distinct poxviruses. From these antigenic proteins, PoxiPred redundantly located up to five epitopes per protein, resulting in 16,817 potential T-cell epitopes which were mostly (i.e., 92%) predicted as being reactive to CD8+ T-cells. PoxiPred is able to, on a single run, identify antigens and T-cell epitopes for poxviruses with one single input, i.e., the proteome file of any poxvirus.

## 1. Introduction

Poxviruses (members of the *Poxviridae* family) are large, complex viruses, with linear double-stranded DNA (dsDNA) genomes ranging from 135 to 360 kbp [1,2]. It has been well-reported that poxviruses’ genomes are entirely replicated in the cytoplasm [2]. The members of this large family exist throughout the world and can cause a plethora of diseases in humans and animals (reptiles, birds, and mammals) [3]. Poxviruses have the capability to spread by aerosol, insects, and direct contact [4]. Based on their genome architecture and evolutionary relationship, the International Committee on Taxonomy of Viruses (ICTV) classified *Poxviridae* members into two subfamilies, *Chordopoxvirinae* (infect vertebrates) and *Entomopoxvirinae* (infect insects). The subfamily *Chordopoxvirinae* is further divided into 18 genera; among them, four genera, Molluscipoxvirus, Orthopoxvirus, Parapoxvirus, and Yatapoxvirus, are known to cause human infections [5,6,7]. As many as half of the conserved genes (around 100 in total) of chordopoxviruses are also found in entomopoxviruses [4]. Based on previous reports, reptiles, birds, and over 30 mammals have been defined as vertebrate hosts of chordopoxviruses [2,4]. To date, the genomes of the many known poxviruses have been sequenced and well annotated with the aid of molecular biology and bioinformatics approaches, and many more genomes will be sequenced and annotated in the upcoming years (https://www.ncbi.nlm.nih.gov/genomes/GenomesGroup.cgi?taxid=10240) (accessed on 1 September 2023). The recognized contagious diseases associated with this subfamily include Mpox (formerly Monkeypox), smallpox, cowpox, and lumpy skin disease, which affects cattle [8]. Variola virus, the causative agent of smallpox, has been classified as the most infamous member of the *poxviridae*. This highly fatal disease was responsible for causing millions of deaths before its successful eradication from the natural environment [9]. 

Today, several poxviruses continue to pose threats to the world, including Mpox (an ongoing pandemic) and lumpy skin disease (LSD) in cattle. Mpox is a viral illness caused by the monkeypox virus and has the capacity to infect a broad range of animals, including humans. This infectious disease can be spread through human-to-human (H2H) transmission, with contaminated materials or with infected animals [8]. The Mpox virus is divided into two clades, I (formerly the Congo Basin Clade) and II (formerly the West African Clade) [10]. The world is experiencing an ongoing pandemic due to the Mpox virus belonging to Clade II, specifically referred to as Clade IIb, as recognized by WHO in 2022 [8]. As of 23 September 2023, a total of 90,168 confirmed cases and 157 deaths in 115 countries with local transmission have been identified [11]. Lumpy skin disease is an acute to chronic, highly infectious skin disease that affects cattle and water buffalo, caused by the lumpy skin disease virus (LSDV) poxvirus. This very important emerging transboundary disease is transmitted by a range of arthropods, including blood-feeding insects and ticks [12]. LSD has emerged in most East European and Asian countries, and is well documented for its large genome structure and its high resistance to environmental conditions [12,13,14]. Other than these two viruses, some other poxviruses, including tanapoxvirus, cowpox virus, and Yaba-like disease virus have the capacity to infect humans and cause morbidity [4,15,16,17,18].

Currently, the proper treatment methods to overcome or treat these infectious diseases caused by poxviruses are still limited; therefore, there is a pressing and urgent demand to develop effective vaccines against poxviruses. Epitopes, also known as antigenic determinants, are defined as the portion of a foreign protein or antigen that could elicit an immune response mediated by antibodies or T- or B-cells. These antigenic determinates offer a targeted approach to vaccines against infectious diseases [19]. Epitope-based vaccine development is one of the most popular applications of immunoinformatics and is widely used for the development of effective vaccines against a plethora of pathogens [20]. It has been reported that epitope-based vaccines offer optimal therapeutic effectiveness with minimal side effects [21]. There are different major challenges for the prediction of peptide-based significant epitopes, and the prediction of T-cell epitopes (TCEs) is one of them [22]. Over the decades, a wide set of direct and indirect methods has been developed for the prediction of TCEs based on sequential and structural analysis and MHC binders [23]. However, there is a still a need for more accurate methods for the development of high-throughput epitopes to accelerate vaccine development efforts against a wide set of diseases. 

Recent advancements in machine learning (ML) methods have enabled the computational biologist–immunologist to design accurate epitopes for vaccine development. In these regards, viral proteins and peptides are commonly coded into numeric features such as Z-descriptors [24], representing their structural conformations in a way such that ML applications can classify proteins/peptides with different functions (i.e., antigens and non-antigens; epitopes and non-epitopes, among others). Moreover, some of the methods that enable immunoinformatics have not yet fully adapted to the arrival of ML and still employ mechanistic statistics as a basis of classification [25], which might fail to capture the distinctive signal portrayed by data with known immune function. We also argue that current methods are still limited to designing genome-specific epitopes to cope with emerging and re-emerging viral diseases. In the present study, we attempted to develop an AI-based method for predicting antigen proteins as well as design T-cell epitopes targeting 25 poxviruses belonging to different genera of the *Chordopoxvirinae* subfamily.

## 2. Materials and Methods

### 2.1. Retrieval of Known T-Cell Epitopes

Our method consists of three instances of prediction: (i) prediction of antigens, (ii) prediction of T-cell epitopes, and (iii) prediction of type of T-cell epitope. First, to predict T-cell epitopes, we gathered a dataset comprising 977 experimentally verified T-cell epitopes [26](Grifoni et al., 2022) originating from various orthopoxviruses, including Variola (69 epitopes), Vaccinia (863 epitopes), Cowpox (1 epitope), Mpox (2 epitopes), and Ectromelia (42 epitopes) viruses. These T-cell epitopes, which constitute the ‘positive’ epitopes exhibit a varied length, spanning from 11 ± 3 amino acids (aa), with a minimum length of 8 aa and a maximum length of 25 aa. To constitute a negative dataset, we obtained 977 random linear peptides within the same protein in which the T-cell epitopes were originally located. This resulted in a 1:1 (ratio positive to negative) balanced dataset for classification. Second, each epitope is annotated as per its reactivity (i.e., CD4+ and CD8+ T-cell epitopes). Our T-cell epitopes consist of 318 CD4 and 659 CD8 epitopes. We used this information to propose another instance of classification of epitopes to predict their T-cell reactivity. Third, to predict antigens, we have isolated the molecular parent (i.e., protein of origin) of each ‘positive’ epitope. This resulted in 217 unique proteins across five orthopoxviruses. We selected an additional set of 217 proteins, which are not the molecular parents of the T-cell epitopes, to constitute a negative dataset of antigens in a ratio of 1:1. 

Finally, for comparison purposes, we obtained an additional set of 1067 T-cell epitopes from IEDB. A total of 2 epitopes for bovine popular stomatitis virus, 3 epitopes for cowpox virus, 43 epitopes for ectromelia virus, 3 epitopes for tanapox virus, 944 epitopes for vaccinia virus, and 73 epitopes for variola virus. These epitopes had their molecular parents tracked to obtain their antigen of origin. We used IEDB’s antigens and epitopes for validating the models with unseen data. After removing duplicates, we were left with 108 unique antigens, 80 unique T-cell epitopes, and 15 unique TCD4+ epitopes. Also, these epitopes were compared with the epitopes predicted by our tool in regard to their length and amino acid composition.

### 2.2. Retrieval of Proteomes 

We obtained proteomes of 25 poxviruses belonging to eight different genera of the subfamily *Chordopoxvirinae*, including Avipoxvirus (Canarypox virus, Fowlpox virus, and Turkeypox virus), Capripoxvirus (Sheeppox virus, Goatpox virus, and Lumpy skin disease virus), Leporipoxvirus (Myxoma virus), Molluscipoxvirus (Molluscum contagiosum virus), Orthopoxvirus (Camelpox virus, Cowpox virus, Ectromelia virus, Horsepox virus, Monkeypox virus, Tetrapox virus, Vaccinia virus, Variola virus, and Volepox virus), Parapoxvirus (Bovine papular stomatitis virus, Orf virus, Pseudopox virus, Squirrel pox virus, and Sealpox virus), Suipoxvirus (Swinepox virus), and Yatapoxvirus (Yaba monkey tumor virus, and Tanapox virus), which were then downloaded from the comprehensive UniProt (https://www.uniprot.org/) (accessed on 5 September 2023) repository in fasta format. A detailed description of the species, proteome accessions, and number of proteins can be found in Appendix A.

### 2.3. Data Preparation

Once all data were collected (i.e., antigenic and non-antigenic proteins, epitopes and non-epitopes), they were processed and prepared with in-house Python (version 3.9.7) scripts. To establish similarities or differences within a protein, such as toxicity (ref), allergenicity (ref), antigenicity (ref), hydrophobicity, molecular size, and polarity between amino acids, can be annotated by Z-descriptors [24]. Moreover, to eliminate the need for alignment, the Auto Cross Covariance (ACC) method [27] was developed to transform Z-descriptor-annotated proteins into same-sized vectors, benefitting quantitative structural activities relationships (QSARs) between peptides with varied length. Additionally, all the independent variables were scaled to the same magnitude with the StandardScaler function implemented by the sklearn.preprocessing library.

### 2.4. Classification Routines

We used different algorithms to classify each dataset. There are three classification tasks considered: (i) antigen prediction; (ii) epitope prediction; (iii) epitope type prediction. First, we used a panel composed of the algorithms Random Forest (RF), Support Vector Machines (SVMs), Logistic Regression (LR), Gradient Boosting (GB), Extreme Gradient Boosting (XGBoost), and K-Nearest Neighbors (KNN); all the algorithms were built in Python using the packages RandomForestClassifier, SVC, LogisticRegression, GradientBoostingClassifier, XGBClassifier, and KNeighborsClassifier, respectively, of the scikit-learn (version 1.3.1) library. All algorithms were initially executed with default parameters. For the sole purpose of model selection, each algorithm underwent a data split, allocating 80% for training and 20% for validation. The algorithm that presented the highest test accuracy, precision, F1 score, and recall metrics was selected. In case the aforementioned algorithms did not yield a test accuracy, precision, recall, and F1 score all above 80%, a more robust classifier, i.e., Deep Learning Artificial Neural Networks (DL-ANNs) were used with the data splits being restarted. If an algorithm of the initial panel showed satisfactory performance, we re-instantiated the model and used GridSearchCV to search for best hyperparameters. We implemented the DL-ANN with the TensorFlow (version 2.12.0) package. To tune the hyperparameters of the DL-ANN, we had our data split in a 10-fold cross-validation; we then manually evaluated the train/test performance of the models by manually iterating over one, two, and three hidden layers, each with 10, 25, and 50 neurons. We stopped adding hidden layers and neurons to the model once it reached a test loss of less than 0.1. We allowed each model to run for a fixed number of training epochs to systematically explore different hidden layer configurations, avoiding premature stopping, and achieving a consistent evaluation of each architecture. Next, we selected these algorithms and conducted hyperparameter tuning by re-instantiating the selected models and submitting them to a 10-fold cross-validation step to isolate the best hyperparameters. Once the hyperparameters were identified, we re-instantiated each model, restarted the 10-fold cross-validation process in which, iteratively, 9/10 of the data is used for training the model, and the remaining 1/10 is reserved for testing the model so that each testing fold is used exactly once. The performance of both train and test splits is measured per its accuracy, precision, recall, and specificity. The models were then evaluated on their capacity for, iteratively, classifying 1/10 of the data in a way that the models did not have prior access to these data. Ultimately, external data are predicted so the generalization capacity of the model is assessed with the validation data. We included the pipeline of the method described in PoxiPred in Figure 1. 

In Figure 1, we describe the classification rationale of PoxiPred, consisting of three independent stages, i.e., model selection, hyperparameter tuning, and model evaluation.

The python codes that performed the whole classification pipeline of PoxiPred as well as the pre-trained ML models ready to be used by third parties are freely available at https://github.com/gustavsganzerla/poxipred.

## 3. Results

### 3.1. Antigenicity Classification

To determine the best algorithm for classifying the antigens, we ran our data split in training/testing subsets through six different classification algorithms (Appendix A). Initially, none of the algorithms could sustain a performance higher than 70% for all the accuracy, precision, recall, and F1 scores. Thus, we opted to carry over the classification with a Deep Learning Artificial Neural Network (DL-ANN). To determine the best DL-ANN architecture, we trained and tested our data per the accuracy, precision, recall, specificity, and loss in the training simulations with one, two, and three hidden layers each with 10, 25, and 50 neurons. By using three hidden layers each with 50 neurons trained over 500 epochs, we achieved a performance of 0.95, 0.99, 0.92, 0.99, and 0.06 of the accuracy, precision, recall, specificity, and loss, respectively (Table 1). Moreover, in Figure 2A, we show the AUC over each step of the 10-fold cross-validation process, which when averaged, resulted in a mean AUC of 0.93 ± 0.13. In Figure 2B, we show the generalization capacity of the antigen prediction model, which correctly labeled 81.48% of the external data.

### 3.2. Epitope Classification 

To determine the best algorithm for classifying the epitopes, we ran our data split in training/testing subsets through six different classification algorithms (Appendix A). From the initial search with six non-Deep Learning algorithms, we report that none of them achieved a satisfactory classification performance as they failed to show a classification report equal to or higher than 70% in the accuracy, precision, recall, and F1 score. Next, we ran the data through a DL-ANN to try to obtain a better performance in classifying the epitopes. We iterated over one, two, and three hidden layers each with 10, 25, and 50 neurons over 100 learning epochs (Table 2). We report the best classification performance as being three hidden layers with 50 neurons each as it resulted in a test accuracy, specificity, recall, precision, and loss of 0.93, 0.99, 0.86, 0.99, and 0.09, respectively. Moreover, in Figure 3A, we show the AUC over each step of the 10-fold cross-validation process, which when averaged, resulted in a mean AUC of 0.96 ± 0.09. In Figure 3B, we show the generalization capacity of the antigen prediction model, which correctly labeled 73.75% of the external data.

Once our model successfully distinguished between epitopes and non-epitopes, we added another layer of classification regarding the nature of the epitope per T-cell reactivity (i.e., CD4 and CD8 T-cell epitopes). We first determined the best algorithm for conveying the classification (Table 3).

From the six tested algorithms, we selected XGBoost as the best performer. Moreover, we tuned the hyperparameters of the XGBoost classification with 50, 100, 200, and 300 as the number of estimators; 0.01. 0.1, 0.2, and 0.3 as the learning rate; 3, 5, 7, and 9 as the maximum depth; 1, 3, 5 as the minimum sum of hessian weight (minimum child weight) needed in a child; and 0, 0.1, and 0.2 as gamma (i.e., minimum loss reduction to make a further partition of a leaf node of the tree). We report that by setting the number of estimators, learning rate, maximum depth, minimum child weight, and gamma, respectively, as 50, 0.3, 5, 1, and 0.2, the model achieved an accuracy of 0.83. 

With the selected hyperparameters, we further classified the input data in a 10-fold cross-validation step. As the classification is unbalanced (2.07 CD8+ epitopes per 1 CD4+ epitope), we assessed each fold performance per the F1 score, balanced accuracy, geometric mean, and AUC. Respectively, we report the classification metrics of 0.87 ± 0.04, 0.82 ± 0.06, 0.82 ± 0.07, and 0.91 ± 0.03 (Table 4 and Figure 4A). We also obtained an additional 15 independent CD8+ epitopes that are not present in the train/test data. Our epitope type prediction model correctly labeled 11 epitopes as CD8+, totaling 73.33% of the data (Figure 4B). 

### 3.3. Antigen and Epitope Prediction in the Proteome Files of 25 Poxviruses

We obtained a dataset consisting of the proteome files of 25 distinct poxviruses, encompassing a total of 4471 unique proteins. Our goal was to identify potential antigens, uncover potential T-cell epitopes, and distinguish these epitopes in terms of CD4 and CD8 recognition. To achieve this, we implemented the pipeline that employed our classifiers. First, we subjected each of the 4471 proteins individually to our antigen predictor. This initial step yielded a total of 3198 proteins that were predicted as antigens. Subsequently, we conducted epitope searches for each of these positively identified antigens. The size of the predicted epitopes was determined following a distribution with a mean and standard deviation extracted from our training data (i.e., an average size of 11 ± 3 amino acids). Starting from the N-terminus of each antigen, we systematically sliced peptide sequences for submission to the epitope predictor. To prevent exhaustive searches, we set a maximum number of attempts based on the protein’s length divided by the size of the largest epitope observed in our training data range (i.e., 14 amino acids). The search process halted either upon reaching the maximum number of attempts or upon discovering a total of five epitopes per protein. Lastly, every predicted epitope underwent the epitope type filter, resulting in the assignment of a CD4 or CD8 label to each epitope. In total, 16,817 T-cell epitopes were predicted, where 15,389 are predicted to be recognized by CD8+ T-cells and 1428 by CD4+ T-cells. In Figure 5, we show the breakdown of the predictions achieved by PoxiPred.

In Figure 5, we show a histogram consisting of the submission of the proteome files of 25 different poxviruses applied to the PoxiPred predictor. For each *poxvirus* (y-axis), we show the count of PoxiPred (x-axis) for the total number of proteins in the proteome file (orange), number of proteins predicted as potential antigens (yellow), number of total T-cell epitopes predicted (green), number of TCD4+ epitopes predicted (olive), and number of TCD8+ epitopes predicted (maroon).

The predictions derived from the execution of the PoxiPred pipeline have been made publicly accessible as CSV files, available at https://github.com/gustavsganzerla/poxipred. The structure of these files comprises three columns: (i) a description of the protein of origin; (ii) the amino acid sequence of the epitope; and (iii) the designation of CD4 or CD8, denoting the T-cell reactivity of each epitope.

### 3.4. Comparison of the Predicted with Experimentally Verified Epitopes

In order to compare the epitopes obtained with the PoxiPred method, we selected all 1067 of the T-cell epitopes available at IEDB for the viruses we aimed to predict. In total, the epitopes for seven orthopoxviruses were obtained (See Section 2.2). To identify similarities or differences between the properties of our epitopes, we analyzed the distribution of the length and composition of the amino acids of the epitopes predicted by PoxiPred as well as the experimentally validated epitopes; this information is provided in Figure 6. First, we show the mean of the length of each epitope (Figure 6A,D); we report the lengths being similar to the mean length of PoxiPred’s epitopes, at 11.06 ± 2.05 aa, while the mean length of the experimentally validated epitopes from IEDB is 11.49 ± 4.03 aa. The shortest epitope predicted by PoxiPred was 8 aa in comparison to 6 from IEDB. The lengthiest from PoxiPred was 26 compared to 46 from IEDB. We also report the most common aa in PoxiPred’s epitopes (Figure 6B) and IEDB’s epitopes (Figure 6C). We report both sets of epitopes having a leucine (L) as their most common aa, followed by isoleucine (I), serine (S), valine (V), and asparagine (N). The least common aa of PoxiPred’s predictions are, respectively, tryptophan (W), histidine (H), glutamine (Q), and cysteine (C) while IEDB’s are W, C, H, Q.

## 4. Discussion

In our work, we trained and tested three distinct machine learning algorithms to learn a distinct signal that would distinguish antigens, T-cell epitopes, and the T-cell reactivity of each epitope in the context of poxviruses. Our research has delivered sets of epitopes to be used as part of vaccine constructs in a reverse vaccinology methodology.

T-cells are well known to monitor MHC-bound ligands expressed on the cell type surface throughout the body [28]. MHC ligands known to trigger a T-cell immune response are commonly known as T-cell epitopes. Accurately predicting such epitopes is important for phenotyping, tracking, and stimulating T-cells involved in the immune response against a plethora of infectious diseases, autoimmunity diseases, allergies, cancers, and during transplantation. Due to the high degree of MHC polymorphism and disparity in the data volume at different stages of T-cell epitope generation and presentation within living systems, the accurate prediction of T-cell epitopes remains a challenging task. So far, various methods/algorithms and web servers have been developed to predict T-cell epitopes with the fruitful utilization of quantitative metrics (QMs), artificial neural networks (ANNs), and support vector machine (SVM) approaches [29,30]. Despite the availability of many methods, algorithms, and web servers, a pressing demand to develop the T-cell epitope prediction methods still remains. Existing methods are limited to predicting epitopes for a candidate target in a single run and do not cover proteome-wide epitope prediction. To the best of our knowledge, there is currently no automated method available for the prediction of T-cell epitopes in poxviruses at the whole proteome level. 

Good applications of machine learning (ML), in general, have as their starting point, the use of a well-curated input data, so the patterns discovered by the ML approach are valid [31]. In the context of reverse vaccinology, ML has been gaining attention due to its ability to find hidden relationships in non-linear data structures [19,32,33]. In addition, the ML discovery of immune-relevant properties such as antigens and epitopes might be hindered by the way the input data are codified. ML approaches are known to work with numerical inputs, which at first sight, directly clashes with the human way to recognize the components of a protein/epitope (i.e., amino acids, represented as one out of twenty symbols of the Latin alphabet). In this sense, there have been ways to represent non-numerical data as numerical inputs; for instance, one-hot encoding can successfully convert categorical data into a numeric input and it has had relative success in the discovery of DNA motifs [34] (Choong and Lee, 2017). However, we argue that such methods can generate data in an over-abstracted manner and deviate the input data from its domain origins. In these means, converting data per the structural relationship as we applied with QSAR has the potential to highlight similar biological structures and properties as stated by Doytchinova and Flower, 2007 [25]. When comparing the output of PoxiPred with experimentally validated T-cell epitopes from 25 poxviruses, we noticed that only five viruses (i.e., bovine popular stomatitis virus, cowpox virus, ectromelia virus, vaccinia virus, and variola virus) had epitopes validated for. Still, there are 20 poxviruses we considered with no epitopes found in the literature. We attribute this to the fact that some of these viruses might not be widely studied. Thus, we see a gap in epitopes for ‘less popular’ viruses that might be filled by AI-based methods.

The current iteration of PoxiPred is limited to the discovery of T-cell epitopes in the context of *Poxviruses* as the input data we had available at the developmental stage of PoxiPred did not enable us to seek for B-cell epitopes. Moreover, there are characteristics of an epitope such as its toxicity and allergenicity that have not been explored by the current iteration of PoxiPred. In addition, if interested users want to use PoxiPred for the discovery of epitopes for specific *Poxviruses* strands, they would be required to run the freely available code and models of PoxiPred in their own computer environments, which might not be user friendly. We argue that due to the iterative nature of our tool, we intend to release future builds that function as webservices and will allow for the discovery of other immunological properties in other viral families, increasing the scope of PoxiPred.

In conclusion, we were able to convert protein and epitope information in a distinctive signal that was successfully captured by machine learning algorithms and allowed for the prediction of antigens and T-cell epitopes (as well as their T-cell reactivity) in 25 distinct *Poxviruses*. To this extent, we delivered a novel set of epitopes to be further explored in reverse vaccinology designs. 

## 5. Conclusions 

Here, we developed an artificial intelligence-based method to enable the prediction of both antigens and T-cell epitopes in poxviruses, i.e., PoxiPred. Our validation step has achieved T-cell epitopes that are structurally comparable to experimentally obtained epitopes. For these reasons, PoxiPred is a natural evolution to the field of epitope prediction, as it employs Deep Learning, a mathematically robust classifier. The outcomes we obtained with executing ProxiPred as well as the pre-trained models are made publicly available for interested users who want to expand the immune factors around poxviruses and curate information for reverse vaccinology.

## Figures and Tables

**Figure 1 biology-13-00125-f001:**
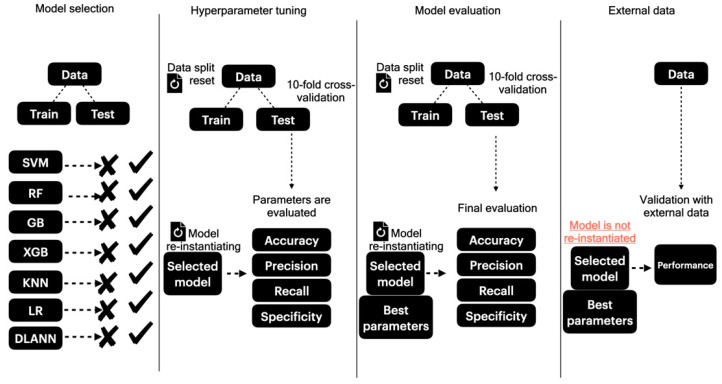
Methodological approach of the classification rationale of PoxiPred.

**Figure 2 biology-13-00125-f002:**
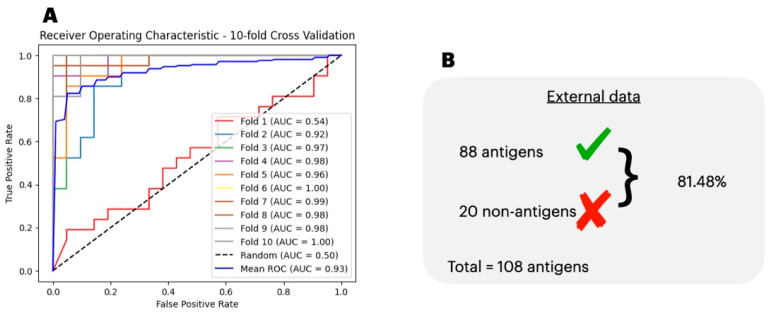
AUC for antigenicity. (**A**), we show the test Area Under the Curve (AUC) score for each fold of a Deep Learning Artificial Neural Network classifying between antigens and non-antigen data. The mean AUC score was 0.93 ± 0.13. (**B**), we show the performance of PoxiPred’s antigen predictor in identifying an external dataset of known antigenic proteins.

**Figure 3 biology-13-00125-f003:**
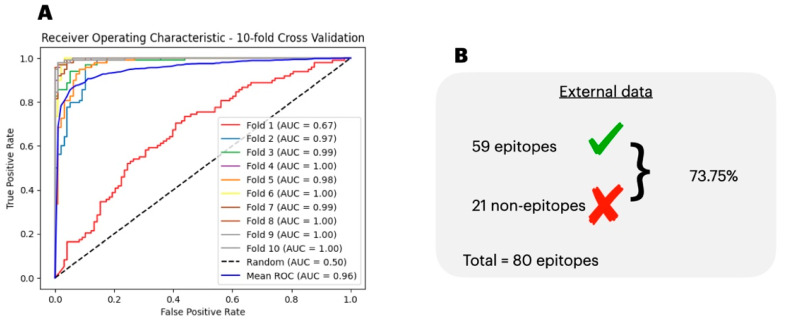
AUC for T-cell epitope classification. (**A**), we show the test Area Under the Curve (AUC) score for each fold of a Deep Learning Artificial Neural Network classifying between antigens and non-antigen data. The mean AUC score was 0.96 ± 0.09. (**B**), we show the performance of PoxiPred’s T-cell epitope predictor in identifying an external dataset of known T-cell epitopes.

**Figure 4 biology-13-00125-f004:**
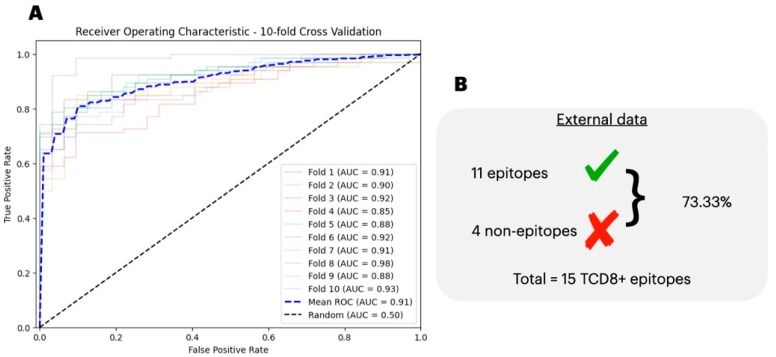
AUC for CD4+ and CD8+ epitope classification in a 10-fold cross validation step over the test folds. (**A**), we show the test Area Under the Curve (AUC) score of an Extreme Gradient Boosting (XGBoost) classification between the CD4+ and CD8+ T-cell epitopes. The mean AUC was 0.91 ± 0.03. (**B**), we show the performance of our model in classifying the independent data.

**Figure 5 biology-13-00125-f005:**
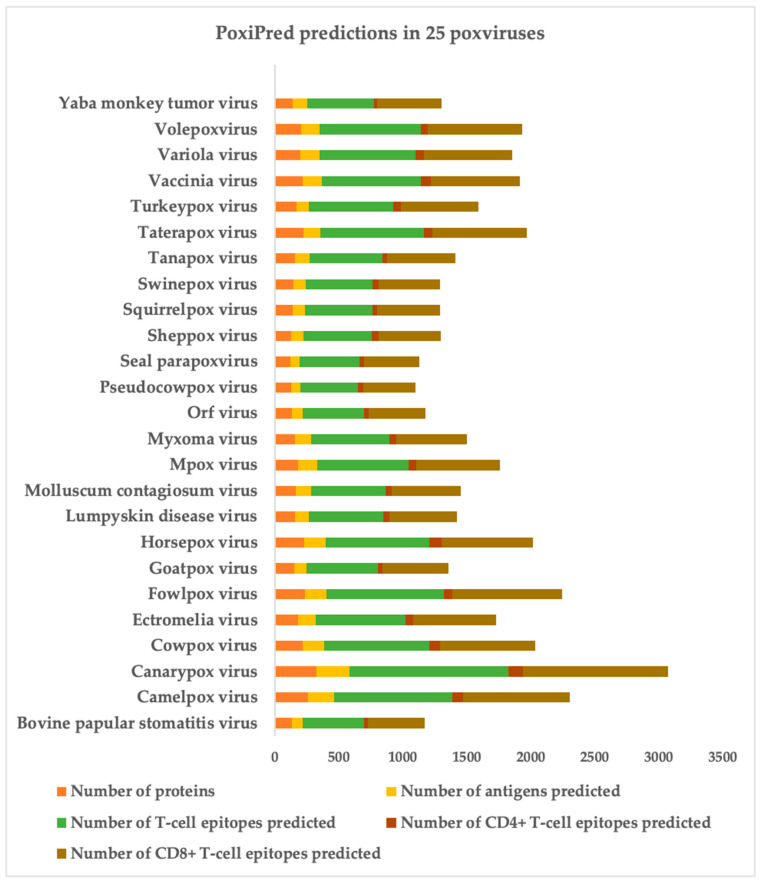
PoxiPred predictions in 25 poxviruses.

**Figure 6 biology-13-00125-f006:**
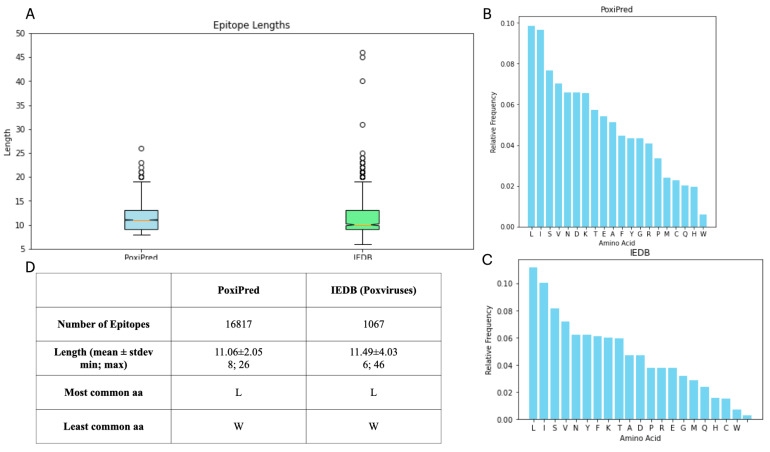
Comparison of the PoxiPred predictions and experimentally validated epitopes of orthopoxviruses. (**A**) shows the length distribution of each set of epitopes, and (**B**,**C**) show the most common amino acids amongst each set of epitopes (PoxiPred’s and IEDB, respectively). Finally, (**D**) summarizes the number of epitopes, and the mean, most, and least common amino acids.

**Table 1 biology-13-00125-t001:** Determining the best Deep Learning Artificial Neural Network architecture to classify antigenic proteins from orthopoxviruses.

	Hidden Layers	Neurons	Accuracy	Specificity	Recall	Precision	Loss	Epochs
Train	1	10	0.542857	1	0.085714	1	0.27382	500
	2		0.697884	1	0.395767	1	0.065661	
	3		0.85873	1	0.71746	1	0.009175	
	1	25	0.661111	1	0.322222	1	0.059489	
	2		0.933598	1	0.867196	1	0.000419	
	3		0.983069	1	0.966138	1	0.000034	
	1	50	0.747354	1	0.494709	1	0.014109	
	2		0.953704	1	0.907407	1	0.000072	
	3		1	1	1	1	0.000001	
Test	1	10	0.533333	0.995238	0.071429	0.4875	0.616637	500
	2		0.680952	0.97619	0.385714	0.871985	0.986844	
	3		0.788095	0.966667	0.609524	0.932477	1.191654	
	1	25	0.65	0.995238	0.304762	0.885714	0.490774	
	2		0.861905	0.985714	0.738095	0.9625	0.364647	
	3		0.840476	0.97619	0.704762	0.958974	0.832513	
	1	50	0.707143	0.990476	0.42381	0.933333	0.30011	
	2		0.883333	0.985714	0.780952	0.957143	0.370044	
	3		0.959524	0.995238	0.92381	0.995	0.066186	

**Table 2 biology-13-00125-t002:** Determining the best Deep Learning Artificial Neural Network architecture to classify T-cell epitopes from orthopoxviruses.

	Hidden Layers	Neurons	Accuracy	Specificity	Recall	Precision	Loss	Epochs
Train	1	10	0.500057	1	0.000114	0.1	0.492214	100
	2		0.506767	1	0.013529	0.7	0.406405	
	3		0.534196	1	0.046385	0.9	0.368531	
	1	25	0.505401	1	0.0108	0.7	0.353444	
	2		0.700928	1	0.401827	1	0.073519	
	3		0.80755	0.999886	0.615183	0.999833	0.028656	
	1	50	0.56396	1	0.127903	0.9	0.163809	
	2		0.900867	1	0.801725	0.00165	0.827208	
	3		0.999943	1	0.999886	1	0.000021	
Test	1	10	0.5	1	0	0	0.621593	100
	2		0.5082	0.998969	0.017473	0.06	0.587974	
	3		0.519487	0.99898	0.040112	0.591667	0.580427	
	1	25	0.503074	0.99898	0.007185	0.55	0.569142	
	2		0.688061	0.998969	0.377341	0.9975	0.395682	
	3		0.765824	0.988776	0.543194	0.940396	0.408892	
	1	50	0.568184	0.998969	0.137513	0.895455	0.41917	
	2		0.827208	0.981633	0.672996	0.960404	0.32027	
	3		0.93145	0.996928	0.86599	0.996532	0.090237	

**Table 3 biology-13-00125-t003:** CD4/CD8 T-cell epitope classification performance.

	Accuracy	Precision	Recall	F1 Score
Random Forest	0.76	0.78	0.92	0.84
Support Vector Machines	0.69	0.69	1	0.82
Logistic Regression	0.65	0.68	0.93	0.79
Gradient Boosting	0.80	0.82	0.81	0.86
Extreme Gradient Boosting	0.82	0.84	0.90	0.87
K-Nearest Neighbors	0.61	0.84	0.54	0.66

**Table 4 biology-13-00125-t004:** Performance of a CD4+ and CD8+ epitope classification in a 10-fold cross validation step over the test folds.

Fold n.	F1 Score	Balanced Accuracy	Geometric Mean
1	0.89	0.72	0.82
2	0.85	0.74	0.72
3	0.88	0.84	0.84
4	0.82	0.71	0.70
5	0.82	0.78	0.78
6	0.90	0.87	0.87
7	0.84	0.79	0.79
8	0.94	0.93	0.93
9	0.89	0.87	0.87
10	0.88	0.84	0.83
Mean	0.87	0.82	0.82
Standard deviation	0.04	0.06	0.07

## Data Availability

All the generated CSV files containing the predictions, the Python code, as well as the pre-trained machine learning models are available at https://github.com/gustavsganzerla/poxipred.

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
