# Peer review of "PoxiPred: An Artificial-Intelligence-Based Method for the Prediction of Potential Antigens and Epitopes to Accelerate Vaccine Development Efforts against Poxviruses"

_biology, 2024, doi:10.3390/biology13020125_

Round 1

Reviewer 1 Report (Previous Reviewer 1)

Comments and Suggestions for Authors

The authors have appropriately corrected the items noted and the paper satisfies the paper's logic adequately.

Author Response

Dear Reviewer, many thanks for encouraging our MS. 

Reviewer 2 Report (New Reviewer)

Comments and Suggestions for Authors

Predicting antigens and epitopes is always an important issue that should be overcome. In this paper, the authors try to use different kinds of machine learning methods to achieve these targets. The author's result shows great performance. However, I have many concerns about these manuscripts, especially the machine learning process. I suggest the author address these issues to improve the paper.

Major concerns:

Generally, to train and evaluate a machine learning model, the dataset is first divided into the training set and the testing set. Then, the training dataset is used for variable selection, hyperparameter selection, and model training. Some of these processes may depend on prediction results, so the training set can be further split into a training set and a validation set. Please note, that the test set should be set aside without being involved in these processes until testing. In this way, the performance will not be overestimated.

However, in this manuscript, the author does not clearly define and correctly use the training, validation, and test set. The test set was not set aside in most stages so the information in the test set has been leaked to the model and all of the results are inaccurate or overestimated. For example, the model selection process is based on the performance of the test set (lines 171-174). So, further comments have to be based on updated results.

Minor concerns

I suggest the author use early stopping strategy for the DL-ANN training.

Please increase the resolution of Figure 5. I cannot see the text of x- or y-axis.

Please provide the versions of the packages such as sklearn.

Comments on the Quality of English Language

Small numbers ranging from one to ten should generally be spelled out (e.g. lines 272)

Round 2

Reviewer 2 Report (New Reviewer)

Comments and Suggestions for Authors

I still have some comments on the machine-learning process:

1. I cannot find Table Sx(S1 S2) in the manuscript, which is mentioned in the manuscript.

2. The author does not address the issue of the test set leaking to the model selection process. In lines 170-172, the authors based on the test set, which is actually treated as a validation set, to select the model. So the performance of the selected model is validation performance rather than test performance. Just like selecting and training the deep learning model, the author should split the dataset into training/validation/test sets, and select models based on the performance of the validation set (early stopping). The performance should be evaluated by the test set. Otherwise, the result is overestimated.

Author Response

Dear Reviewer, 

This manuscript is a resubmission of an earlier submission. The following is a list of the peer review reports and author responses from that submission.

Round 1

Reviewer 1 Report

Comments and Suggestions for Authors

This study is planned to predict the most antigenic epitope by machine learning for vaccine development against poxvirus. In their methodology, the authors appropriately present the design conditions of the model. Concerning the results, the AUC exceeds 0.8 as the model's accuracy, which is considered to meet a certain level of accuracy. On the other hand, a histogram of the epitopes predicted in this study is shown in Figure 4. Still, only a rough number of epitopes are shown, especially for variola virus, which is highly pathogenic to humans, and vaccinia, an existing smallpox vaccine strain. It is necessary to show specific regions for epitopes that are specifically antigenic by machine learning. In particular, for vaccinia, the antigenicity varies greatly depending on the strain type in the preceding review (https://www.mdpi.com/2079-7737/10/11/1158). In this regard, the epitopes obtained in this study need to be compared with the results of studies on smallpox and existing vaccinia strains, which are at least the key to vaccine production.

Reviewer 2 Report

Comments and Suggestions for Authors

Gustavo Sganzerla Martinez et al. present in their manuscript an algorithm and software to find epitopes on a proteome wide level that have a high probability of allowing epitope based vaccine development against viruses of the Poxviridae family using machine learning algorithms. 

Members of the Poxviridae family infect mostly animals but at least two cases are known where a virus has crossed the animal-human barrier, the mpox virus and the lumpy skin causing virus. The variola virus was a member of this family before his official extinction from the biosphere. 

The authors train their software on approx. 800 experimentally verified T-cell epitopes from the proteome of five of these virus. They keep about 200 known T-cell epitopes in reserve to test the quality in prediction of their programme after training. In addition they selected the same number of non-epitope peptides from the same proteins to represent a balanced data set to the training algorithm that contained as many true positive as true negative peptide sequences. 

After adding a deep learning training level of up to 3 levels the programme increased its accuracy in predicting correct antigenic peptide from about 50% to nearly 100% (see table 1 and 2). This shows impressively how such an approach can improve its performance even when the training set is limited. One has to keep in mind that training and test set came from the same organisms. The authors do not make more general claims but limit their programme to the Poxviridae family of virus. 

Ultimately the authors apply their programme to all members of the Poxviridae family and find 16 817 cell epitopes with 15 389 from them being recognised by CD8+ T-cells and 1 428 recognised by CD4+ T-cells. 

I think the data presented in this manuscript support well the claims of the authors. The methods are well described and the programme PoxiPred represents a valid contribution to the bioinformatic toolbox in the vaccine development efforts. 

In my eyes the manuscript can be published as it is. 

Reviewer 3 Report

Comments and Suggestions for Authors

The authors proposed a method to predict the potential T-cell epitopes for multiple poxviruses applaing  an AI-based tool for trained and test antigens and epitopes. However, the weak point of this work is the lack of validation of the method presented both in the results and in the discussion. The authors used experimentally verified data regarding 69 smallpox epitopes, 863 vaccinia epitopes... however, they did not use these results to validate the work data, or it is not clear in the way it is presented. I suggest that the authors write a new topic like "validation of results" to compare the data from this work with more detail with experimental data found in the literature.
